# Spectral Kurtosis of Choi–Williams Distribution and Hidden Markov Model for Gearbox Fault Diagnosis

**Yufei Li [1], Wanqing Song [1], Fei Wu [1,\*], Enrico Zio [2] and Yujin Zhang [1]** 

[1]  School of Electronic & Electrical Engineering, Shanghai University of Science Engineering, Shanghai 201620, China; 15221679101@163.com (Y.L.); swqls@126.com (W.S.); yjzhang@sues.edu.cn (Y.Z.)

[2]  Energy Department, Politecnico di Milano, Via La Masa 34/3, 20156 Milan, Italy; enrico.zio@polimi.it

\*  Correspondence: fei_wu1@163.com

**Abstract:** A combination of spectral kurtosis (SK), based on Choi–Williams distribution (CWD) and hidden Markov models (HMM), accurately identifies initial gearbox failures and diagnoses fault types of gearboxes. First, using the LMD algorithm, five types of gearbox vibration signals are collected and decomposed into several product function (PF) components and the multicomponent signals are decomposed into single-component signals. Then, the kurtosis value of each component is calculated, and the component with the largest kurtosis value is selected for the CWD-SK analysis. According to the calculated CWD-SK value, the characteristics of the initial failure of the gearbox are extracted. This method not only avoids the difficulty of selecting the window function, but also provides original eigenvalues for fault feature classification. In the end, from the CWD-SK characteristic parameters at each characteristic frequency, the characteristic sequence based on CWD-SK is obtained with HMM training and diagnosis. The experimental results show that this method can effectively identify the initial fault characteristics of the gearbox, and also accurately classify the fault characteristics of different degrees.

**Keywords:** Choi–Williams distribution; spectral kurtosis; HMM; gearbox fault classification

## 1. Introduction

In rotating machinery, gearboxes are widely used in various industries as a universal component for changing speed and transmitting power. When the gearbox fails early, the vibration signal usually exhibits nonlinear and nonstationary characteristics [1–4]. If early weak faults are found, and their features are effectively extracted in time, then, equipment maintenance can be performed to reduce the danger [5]. When the gear in the gearbox has local faults such as pitting, spalling, scratching, broken teeth, and spalling of the bearing, it causes a series of transient shock responses. In the early stage of the fault, the shock fault characteristic signal is faint and often obscured by strong background noise. It is especially important to correctly distinguish the working status of the gearbox [6–9]. In recent years, the concept of spectral kurtosis has been proposed to solve the above problems to some extent. Therefore, it is of certain academic significance to study the application of the spectral kurtosis method in gearbox fault diagnosis [10–12].

Currently, the traditional methods of gearbox vibration signal processing and analysis technology mainly include time-domain waveform probability statistical analysis, correlation analysis, coherence analysis, time-domain synchronous averaging, frequency-domain probability statistical analysis, detailed spectrum analysis, etc. The concept of spectral kurtosis (SK) was first proposed by Dwyer [13], who used it to detect transient components in noise signals. The core idea of spectral kurtosis is to be able to calculate the kurtosis value of each spectral line in the frequency spectrum. To further define SK in detail, a short-time Fourier transform of spectral kurtosis (STFT-SK) calculation

method was proposed which helped to link theoretical concepts with practical applications [14]. Then, it was verified that it detected transient signals in nonstationary signals with additive noise. By locating transients in a frequency domain with heavy noise, an improved SK was proposed for early fault detection of bearings. Although kurtosis is based on temporal signals, which is effective under some conditions. Its performance is low in the presence of a low signal-to-noise ratio and non-Gaussian noise [15].

Then, SK was combined with a filter to extract early fault signals. An optimal denoising filter was proposed which enhanced the transient component of the gear vibration signal [16]. Then, the SK-based filtered residual signal was proposed and the local power was defined as the smoothed squared envelope [17]. In 2011, a real-time gear fault feature extraction method was proposed, which combined a one-dimensional map and a band-pass filter, however, they were inherently slow and not suitable for real-time applications [18]. A gear tooth fault detection method based on the maximum correlation kurtosis deconvolution method was proposed. The experimental data were from a gearbox with gear chip fault, and the results were compared between healthy and faulty vibrations [19]. Compared with SK-based gearbox fault feature extraction method, a sparse signal decomposition based on tunable Q-factor wavelet transform (TQWT) was proposed. It showed that the proposed method outperformed empirical mode decomposition and SK in extracting fault features of gearboxes [20].

There are many algorithms that have been combined with SK. Several calculation methods which have been based on SK are summarized, such as those based on STFT, wavelet transform (WT), and Wigner–Ville distribution (WVD) [21]. Among them, STFT-SK is limited by the choice of window function, and poor time-frequency resolution appears in strong noise. WT- SK has better time-frequency resolution, but wavelet bases and decomposition scales are difficult to confirm, and therefore it is not able to obtain optimal diagnosis. WVD-SK has good time-frequency resolution, but there are cross-interference terms that cannot be eliminated.

For failure mode recognition, the current failure mode recognition methods mainly include Bayesian classification, Fisher criterion, nearest neighbor method, fuzzy classification algorithm, neural network algorithm, kernel-based classification algorithm, etc. [22–24]. However, these methods have been based on static analysis, ignoring the dynamic information of gearbox failure changes. Among them, the neural network model is the most widely used, but the neural network model mainly deals with the static classification process and is not suitable for the dynamic signal processing of gearbox failures [25,26]. In addition, the neural network model requires large samples. When the samples are limited, it shows poor generalization ability and the possibility of the optimization process is easily trapped in local extremes [27].

The hidden Markov model (HMM) is a statistical analysis model used to describe a Markov process with hidden unknown parameters. The HMM model has strong feature classification capabilities based on these two stochastic processes. It is especially suitable for statistical analysis of nonstationary, repetitively poor dynamic signals. According to the pattern matching principle, the HMM model trains a reliable model with a small number of samples and finds the pattern most similar to the detection signal as the recognition result [28]. The HMM model has been well applied in the fields of speech recognition, traffic monitoring system, and medical image recognition [29–31]. When compared with neural networks, it retains more statistical information of training data and has a higher recognition rate and robustness [32].

Therefore, in this paper, the combination of LMD, Choi–Williams distribution and SK (CWD-SK), and HMM are used to identify the initial failure of the gearbox and accurately distinguished the four types of failure. Then, the extracted CWD-SK is used as the feature vector to input HMM and back propagation (BP) neural network to compare the accuracy of fault diagnosis.

This paper is organized as follows: In Section 2, the main calculation steps of CWD-SK and the impact of window functions are described; in Section 3 the basic principles of HMM are outlined; in Section 4, the application of this method to gearbox fault diagnosis is introduced; and in Section 5, the summary is stated.

## 2. SK Based on CWD

### 2.1. Definition of SK

SK detects signals to non-Gaussian signals, and also determines the frequency of the excited component. For the time being, it is difficult to describe nonstationary signals. We use the Cramer–Wold decomposition to describe a stationary random process in the time domain. We define signal $Y(t)$, as the response of the system with time varying impulse response $h(t, s)$, excited by a signal $X(t)$. Then, $Y(t)$ is presented as

$$Y(t) = \int_{-\infty}^{+\infty} e^{2\pi f t} H(t, f) dH(f) \tag{1}$$

where, $H(t, f)$ is the time varying transfer function of the considered system and is interpreted as the complex envelope of the signal $Y(t)$ at frequency $f$. SK is based on the fourth-order spectral cumulant of a conditionally nonstationary process (CNS) process:

$$C_{4Y}(f) = S_{4Y}(f) - 2S_{2Y}^2(f) \quad (f \neq 0) \tag{2}$$

where, $S_{2n}(f)$ is the second-order spectral cumulant, which is the measure of the energy of the complex envelope, it can be expressed as Equation (3):

$$S_{2nY}(t, f) = E\left\{\left|H(t, f)dX(f)\right|^{2n}\right\}/df = E\left\{\left|H(t, f)\right|^{2n}\right\} \cdot S_{2nX} \tag{3}$$

Therefore, the SK is defined as the energy normalized cumulant, which is a measure of the peak of the probability density function H:

$$K_Y(f) = \frac{C_{4Y}(f)}{S_{2Y}^2(f)} = \frac{S_{4Y}(f)}{S_{2Y}^2(f)} - 2 \quad (f \neq 0) \tag{4}$$

### 2.2. Algorithm of CWD-SK

For nonstationary signal $x(t)$, its instantaneous autocorrelation function is expressed as:

$$R(t, \tau) = x(t + \frac{\tau}{2})x^*(t - \frac{\tau}{2}) \tag{5}$$

The Fourier transform of $R(t, \tau)$ is the WVD of the signal $x(t)$:

$$X_{WVD}(t, f) = \int_{-\infty}^{+\infty} x(t + \frac{\tau}{2})x^*(t - \frac{\tau}{2})e^{-jf\tau}d\tau \tag{6}$$

In order to eliminate the existence of WVD cross-interference terms, the following window function (also called exponential kernel function) is derived

$$g(\tau, v) = \exp(-a\tau^2 v^2) \tag{7}$$

The inverse Fourier transform of the exponential kernel function is expressed as:

$$G(t, \tau) = \int_{-\infty}^{+\infty} g(\tau, v)e^{jvt}dv = \sqrt{\frac{\sigma}{4\pi\tau^2}} \exp(-\frac{\sigma t^2}{4\tau^2}) \tag{8}$$

Now, $g(0, \tau) = g(\tau, 0) = 1$, $g(0, 0) = 1$, and when $\tau \neq 0$, $g(\tau, v) < 1$, where $\tau$ is the time shift parameter. $C_x(t, f)$ is expressed as:

$$C_x(t, f) = \iint \sqrt{\frac{\sigma}{4\pi\tau^2}} \exp(-\frac{\sigma t^2}{4\tau^2})x(\mu + \frac{\tau}{2})x^*(\mu - \frac{\tau}{2})e^{-j2\pi f\tau}d\mu d\tau \tag{9}$$

where $\theta$ is offset parameter, $\sigma$ is scale factor and usually constant. If the value of $\sigma$ is too large, the resolution of the self-term is higher; if the value of $\sigma$ is too small, the suppression performance of the cross term is better. In general, the choice of $\sigma$ needs to consider both the self-term resolution and the cross-term suppression. Therefore, the kernel function ensures higher time-frequency resolution, and also effectively suppresses the cross-terms of two functions with different frequencies and time centers.

According to the definition of CWD, the second-order instantaneous spectral distance and fourth-order instantaneous spectral distance of $x(t)$ are obtained as follows:

$$
\begin{cases}
\hat{S}_{2x}(f) = E\left\{\left|C_x(t,f)\right|^2\right\}_k \\
\hat{S}_{4x}(f) = E\left\{\left|C_x(t,f)\right|^4\right\}_k
\end{cases}
\tag{10}
$$

where, $E\{\bullet\}_k$ represents the average of $k-th$ order. $k$ represents the number of sampling points.

Finally, Equation (9) brings in Equation (4), and CWD-SK algorithm is defined as follows:

$$
k_x(f) = \frac{C_{4x}(f)}{\hat{S}_{2x}^2(f)} = \frac{\hat{S}_{4x}^2(f) - 2\hat{S}_{2x}^2(f)}{\hat{S}_{2x}^2(f)} = \frac{\hat{S}_{4x}^2(f)}{\hat{S}_{2x}^2(f)} - 2, (f \neq 0)
\tag{11}
$$

### 2.3. Impact of Window Functions

SK is a time-frequency analysis algorithm whose time-frequency resolution is closely related to the window function [14]. Some examples of window functions include rectangular, Hanning, Hamming, Blackman and Kaiser windows, etc. When the fluctuation is too large, we usually think that the value of $k$ increases, which affects the end result. Therefore, it is necessary to compare the smoothness of the curves.

In this type, we use curvature to intuitively compare the smoothness of CWD-SK for different window functions. Usually assuming a curve $y = f(x)$ exists, the magnitude of the curvature value is calculated by Equation (12).

$$
K = \frac{\left|f''(x)\right|}{\left(1 + f'^2(x)\right)^{3/2}}
\tag{12}
$$

$f'(x)$ represents the first derivation of $x$, $f''(x)$ represents the second derivation of $x$.

In order to evaluate the complexity of the curve shape, a curvature deviation is proposed [33]. Here the smoothness of the curve is determined in Equation (13).

$$
Smoothness = \sum_{n=1}^{m} abs(k_n - k_{mean})
\tag{13}
$$

where $n$ represents the index of sampling point, $m$ is the length of data, $k_n$ represents curvature of the $n-th$ discrete point, and $k_{mean}$ represents the average of the estimated curvature, respectively.

## 3. Diagnosis Flow Based on HMM

For the fault characteristic indexes that have been extracted to different degrees, first, the normalization and quantification of the characteristic indexes are performed. Secondly, to establish the HMM model, the obtained observation sequence is set to a finite discrete value. Finally, the discretized value is used as the quantized model training feature value.

The hidden Markov models are based on Markov chains, there are N states in the system, such as $S = \{S_1, S_2, \ldots, S_n\}$, and called $q_t$ at time t. The transition matrix between states is $A = \{a_{ij}\}$.

$$
a_{ij}(k) = P\left[q_{i+1} = S_j | q_t = S_i\right], 1 \leq i, j \leq N
\tag{14}
$$

For some HMMs, any state can reach other states in one transition; in other HMMs, when $a_{ij} > 0$, it transitions between certain states can occur.

The difference between HMM and Markov chain is that for each state, the outside world can only make one observation and obtain an n-dimensional observation vector $V_k$. This vector is related to the state of the system and is discrete or continuous.

For continuously distributed observations, the probability that the state $j$ corresponds to the observation vector distribution is $B = b_j(v_i)$, here

$$b_j(v_i) = \mathrm{P}\big[v_i \big| q_t = S_j\big], 1 \le j \le N \tag{15}$$

The probability distribution is generally taken as a mixed Gaussian distribution:

$$b_j(v_i) = \sum_{m=1}^{M} w_{j,m} N(o_t, \mu_{j,m}, \sum\nolimits_{j,m}) \tag{16}$$

where $M$ is the number of mixed Gaussian distributions, $w_m$ is the positive mixing weight, the sum is 1, and $N(o_t, \mu_{j,m}, \sum_{j,m})$ is an n-dimensional Gaussian distribution.

The initial state distribution is $\pi = \{\pi_i\}$, where

$$\pi_i = P[q_1 = S_i], 1 \le i \le N \tag{17}$$

So far, the parameters of the HMM are summarized as three groups $\lambda = (A, B, \pi)$. The observation sequence generated by this model is $O = o_1 o_2 \ldots o_T$, $o_t$ is the observation vector at time t, and T is the total observation length.

Figure 1 shows the diagnosis process of HMM. The Baum–Welch algorithm [34,35] is suitable for training, adjusting, and optimizing parameters in observation sequences. Therefore, the obtained sequence of observation probability values is similar to the sequence of observation values. Then, we calculate the maximum value, HMM status recognition, and the status of different fault levels. The purpose is to establish the corresponding HMM model, get the date of the unknown fault state and, then, enter each model in turn, and calculate and compare the possibilities. Finally, the unknown signal failure type is obtained from the output probability of the maximum model. As can be seen, the most probable path through the sequence is observed by the Viterbi algorithm. The above is the HMM diagnosis process.

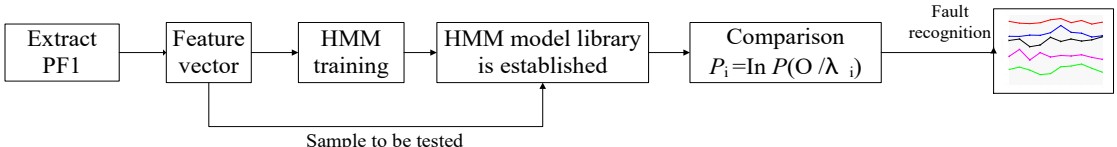

**Figure 1.** Diagnosis flow based on hidden Markov models (HMMs).

## 4. Experimental Data Analysis

### 4.1. Experiment Platform and Data Preprocessing

All the experimental data are obtained from the MFD310 system, as shown in Figure 2. This gearbox (reducer) is a three-stage transmission, consisting of four shafts, four pairs of bearings, seven straight gear teeth, and a box. The power source of this system is the motor. The input shaft of the first gear box is first connected to the first speed torque sensor and, then, connected to the motor. After the first gear box is decelerated, the output shaft and the input shaft of the second gear box are connected. The output shaft is connected to the second speed torque sensor after deceleration of the second gear box and, finally, connected to the eddy current brake. So that the location of the fault is relatively close and the required data can be accurately and sensitively collected, the vibration signals are collected

respectively in five states of the gearbox, such as normal, slight fault, moderate fault, severe fault, and flaking. The frequency of shaft II is 7 Hz, the frequency of shaft I is 10 Hz, the sampling points is 4096, and the sampling frequency is 3387.77 Hz. The gearbox meshing frequency is 307 Hz.

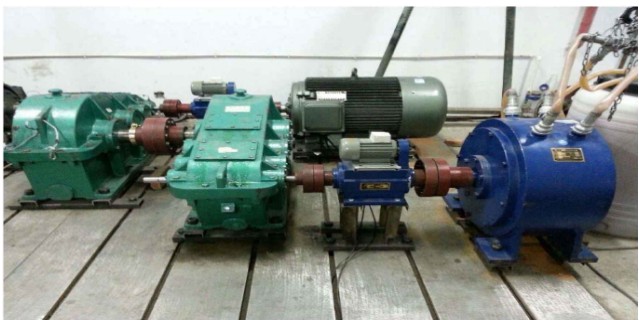

**Figure 2.** MFD310 gearbox experimental platform for various working conditions fault diagnosis.

Five kinds of vibration signals for the same gearbox are collected respectively, and the time domain and Fast Fourier transform analysis are performed, which is chaotic, complicated. It shows a random distribution. Therefore it is almost impossible to judge the running state of the gearbox based on this. Therefore, this paper chooses the adaptive time analysis, Fast Fourier transform (FFT) analysis, and Local mean decomposition (LMD) to perform data preprocessing on the five states of the gearbox (as shown in the Figure 3). LMD preprocesses the vibration signals in the five states of the gearbox to remove interference factors such as noise.

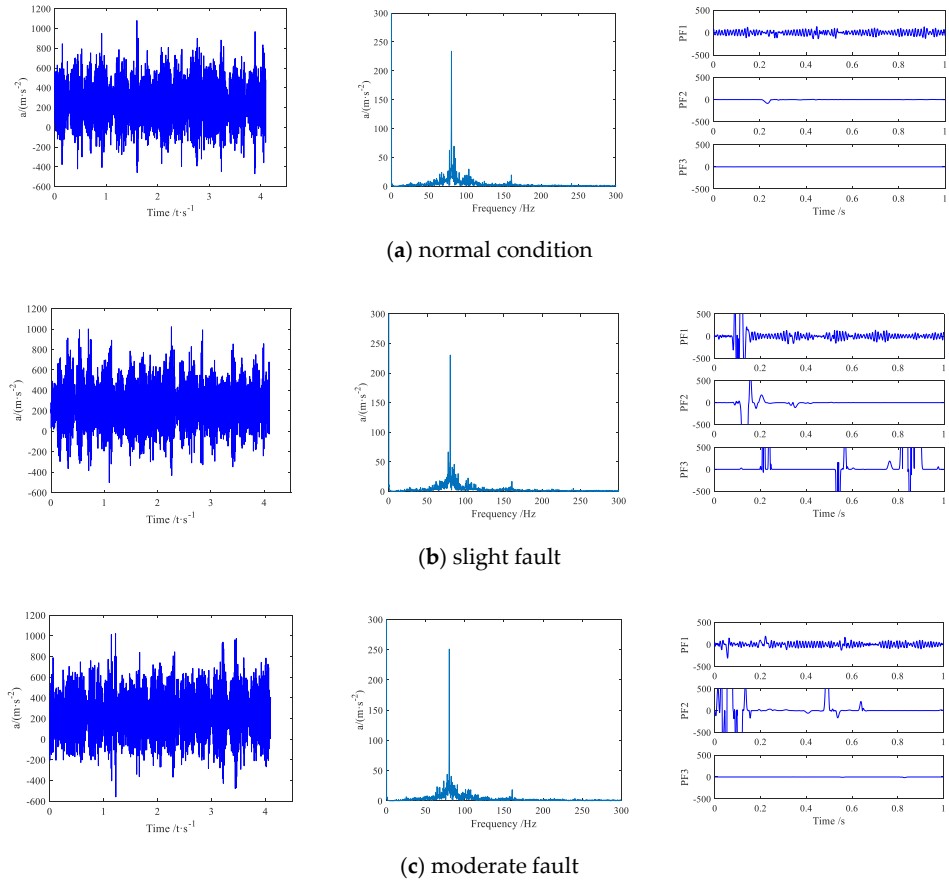

(**a**) normal condition

(**b**) slight fault

(**c**) moderate fault

**Figure 3.** *Cont*.

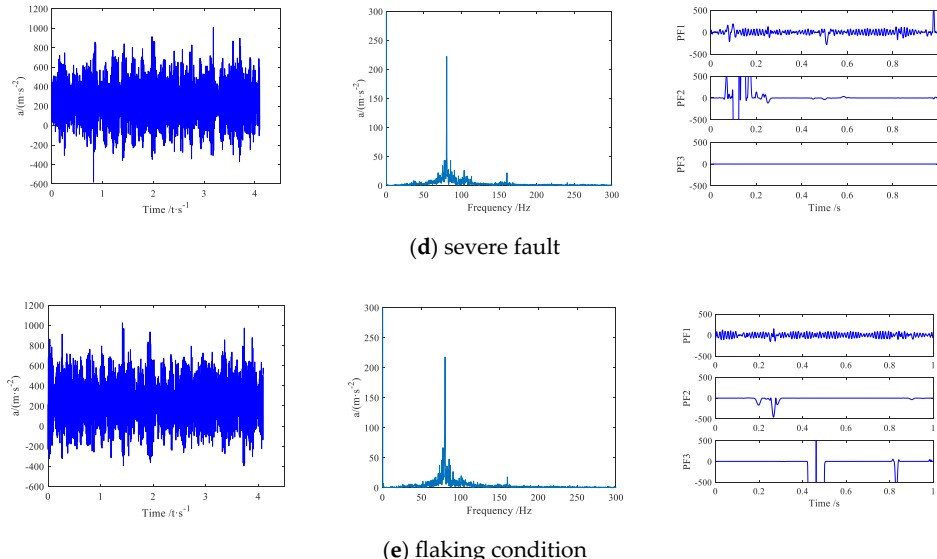

(**d**) severe fault

(**e**) flaking condition

**Figure 3.** Time domain, FFT and LMD plots in 5 states.

For the calculation of each PF component, the corresponding kurtosis value is calculated (in Table 1), and the PF with the larger kurtosis value is selected (for the convenience of subsequent CWD-SK analysis, usually the PF with the highest kurtosis is selected) for the CWD-SK processing.

**Table 1.** PF Kurtosis value of the five gearbox signals.

| PF | 1 | 2 | 3 |
|---|---|---|---|
| Kurtosis(normal) | 3.730 | 3.212 | 3.098 |
| Kurtosis (slight fault) | 3.980 | 3.402 | 3.196 |
| Kurtosis (moderate fault) | 4.645 | 4.320 | 4.089 |
| Kurtosis (severe fault) | 4.966 | 4.781 | 4.342 |
| Kurtosis(flaking) | 5.905 | 5.455 | 5.403 |

### 4.2. Initial Fault Feature Extraction Based on CWD-SK

SK is a statistical tool. The non-Gaussian component in the signal is detected by SK. And the existence of the transient and its position in the frequency domain can be clearly pointed out. Once the gearbox fails, the frequency component of the faulty gear increases as compared with the normal state, so the amplitude of SK in the frequency increases accordingly.

The CWD-SK in PF1 is calculated according to Equation (11). This method calculates the value based on CWD-SK, which is denoted as *k*. The value of CWD-SK in normal conditions is shown in Figure 4a. In normal conditions, $K_{1mean}$ is 1.75 in the frequency domain. When the gearbox fails early, $K_{2mean}$ becomes 2.04. When the gearbox fails, the K value is generally greater than two. In other words, there is a slight failure of the gearbox, Figure 4b, which is in line with the actual situation. In the case of moderate and severe gearbox failures, $K_{3mean}$ and $K_{4mean}$ are almost the same and cannot be distinguished according to conventional methods. The HMM model mentioned below is required for further training and classification. Thus, CWD-SK can identify initial gearbox failures accurately.

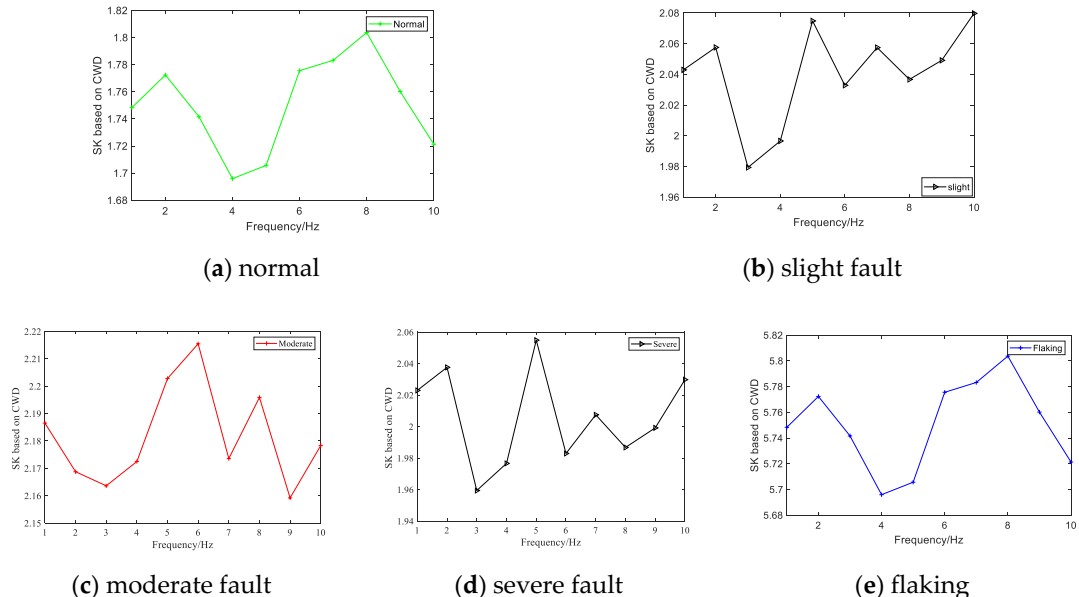

**Figure 4.** Value of Choi–Williams distribution and spectral kurtosis (CWD-SK) with five cases.

### 4.3. Selection of Window Function

As shown in Table 2, in order to avoid the sudden increase and decrease of the SK curve, the average value of CWD-SK is selected as the characteristic value. Under normal circumstances, the average CWD-SK of the gearbox is 1.75, which is in line with the analysis that SK is not more than two in the case of normal condition. When the gearbox has a slight failure, the average value of CWD-SK obviously increases to 2.023. This indicates that CWD-SK detects the initial failure well, which is consistent with the actual situation.

**Table 2.** The mean of CWD-SK in five cases.

| Condition | The Mean of CWD-SK |
|-----------|--------------------|
| Normal    | 1.756              |
| Slight    | 2.023              |
| Moderate  | 2.187              |
| Severe    | 2.019              |
| Flaking   | 5.746              |

From Equation (9), when the value is too large, it proves that the smoothing effect of this curve is not ideal. If not, it can prove that the curve smoothing effect is better. According to Table 3, four typical window functions are selected. It is obtained that CWD-SK is not sensitive to the selection of the window function. It reduces the difficulty of selecting the window function.

**Table 3.** Average smoothness of SK with different window functions.

| Window Functions | Smoothness |
|------------------|------------|
| Rectangular      | 0.447      |
| Hanning          | 0.449      |
| Hamming          | 0.508      |
| Blackman         | 0.458      |

### 4.4. Five Types of Gear Fault Characteristics Classification

We can see that $K_{2mean}$ is larger than $K_{1mean}$, so we can identify initial gearbox failure and the slight faults and the normal are well distinguished. When the gearbox begins to appear moderate and severe

fault, the average value of *K* cannot be obvious distinguished. Therefore, it is necessary to use HMM for fault status identification.

The training curve is shown in Figure 5. The HMM modeling and training, slight fault, moderate fault, severe fault, and flaking represent four kinds of hidden state recorded as $\lambda_2$, $\lambda_3$, $\lambda_4$, $\lambda_5$ states, respectively, in the gearbox. The initial probability distribution vector B$\lambda$, the initial state transition matrix $\pi\lambda$, and the initial observation probability matrix are all obtained by random functions and, then, go to zero and normalize to a $10 \times 4$ matrix.

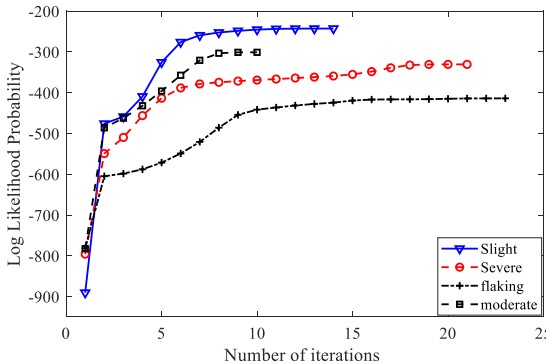

**Figure 5.** Log-probabilities output five fault signals at various HMM model.

In order to prevent situations where the model is caught in a wireless loop or training fails, the Lloyd algorithm is used to scalar quantize the CWD-SK sequence, and input the quantized sequence into the HMM. The training algorithm uses the Baum–Welch algorithm. As the number of iterations increases, the maximum log-likelihood estimate increases until convergence is reached. After training, four hidden HMM recognition models are obtained.

A total of 20 groups (five groups of each state) are selected based on the CWD-SK feature vector as training samples. Before training, the Lloyd algorithm is used to scalar the feature vector and the Baum–Welch algorithm is used for training. During the HMM training process, as the number of iterations increases, the maximum natural logarithmic estimate value continues to increase until it reaches convergence. After training, the HMM recognition models corresponding to four hidden states are obtained. All states reach convergence when the number of iterations is 20, and the convergence speed is fast.

After the HMM model training is completed, a classifier based on the fault state of the gearbox is established. As can be seen, the four fault states of the gearbox reach rapid convergence after 20 iterations. For the trained HMM model, the remaining (at $\lambda_2\lambda_3\lambda_4$ and $\lambda_5$ model) 20 groups of CWD-SK feature vectors (five states of each state) are input as test samples. Of course, before testing, it uses the Lloyd algorithm to scale the CWD-SK sequence and inputs the scaled quantized sample CWD-SK sequence into each state.

According to Table 4, the higher the degree of failure of the second same type of failure, the larger the log-likelihood estimate. At the same time, the output value of the natural logarithmic probability estimation of each state is the maximum in this state. For the four failure states of the gearbox, diagnosis results show that this method can accurately classify faults. From the experimental data, the HMM fault diagnosis model can successfully identify the four fault states of the gearbox with high accuracy and a small amount of data.

**Table 4.** HMM recognition result.

| Fault Case | Logarithm Likelihood Probabilities of the Input Sample Model | | | | |
|---|---|---|---|---|---|
| | $\lambda_2$ | $\lambda_3$ | $\lambda_4$ | $\lambda_5$ | Recognition Result |
| Slight fault | −15.114 | −∞ | −∞ | −∞ | $\lambda_2$ |
| Moderate fault | −54.124 | -55.964 | −∞ | −∞ | $\lambda_3$ |
| Severe fault | −∞ | −62.14 | −76.44 | −134.82 | $\lambda_4$ |
| Flaking | −∞ | −132.67 | −199.211 | −211.342 | $\lambda_5$ |

As shown in Table 5, in order to prove the accuracy of CWD-SK as the feature vector, HMM is input for classification and recognition. Comparing the training results of HMM with the training results of the BP neural network, we set some parameters of BP as follows: trainParam_Show = 10, trainParam_Epochs = 1000, trainParam_mc = 0.75, trainParam_Lr = 0.05, trainParam_lrinc = 1.5, and trainParam_Goal = 0.1. For SK as compared with the input of feature vectors, the overall recognition accuracy of the CWD-SK model is higher.

**Table 5.** Comparison of CWD-SK for fault recognition.

| Recognition Model | | Slight Fault | Moderate Fault | Severe Fault | Flaking | Recognition Rate |
|---|---|---|---|---|---|---|
| CWD-SK | HMM | 5 | 4 | 5 | 5 | 95% |
| | BP | 4 | 4 | 5 | 5 | 90% |
| SK | HMM | 4 | 5 | 4 | 5 | 90% |
| | BP | 4 | 4 | 4 | 5 | 85% |

## 5. Conclusions

In this study, we focus on the nonlinear and nonstationary characteristics of a gearbox's five state (normal, slight fault, moderate fault, severe fault, and flaking) of vibration signals, and a gear fault extraction and classification recognition method, CWD-SK, is combined with HMM. CWD-SK is insensitive to window function types and anti-noise, which avoids the difficulty of selecting the window function. After LMD decomposition, a PF component with a larger kurtosis value is selected according to the maximum kurtosis criterion. Then, the average value of the CWD-SK is further calculated, and the characteristic frequencies of the gearbox in normal and initial slight fault conditions are extracted. Finally, the HMM is used to identify the fault pattern of the characteristic signal after extracting feature vectors and normalizing the signal. The experimental results prove that the method can identify initial gearbox failures and accurately classify the different fault status of the signals.

**Author Contributions:** Conceptualization, Y.L. and W.S.; Data curation, Y.L., W.S. and F.W.; Formal analysis, Y.L., W.S. and F.W.; Funding acquisition, F.W., Y.Z. and W.S.; Investigation, Y.L., W.S. and F.W.; Methodology, Y.L. and W.S.; Project administration, W.S., F.W. and Y.Z.; Resources, W.S., F.W. and E.Z.; Visualization, W.S., F.W. and Y.Z.; Writing–Original draft, Y.L.; Writing–Review & editing, Y.L., W.S., F.W., E.Z. and Y.Z. All authors have read and agreed to the published version of the manuscript.

**Funding:** This project was funded by the Key Project of Science and Technology Commission of Shanghai Municipality (Grant No. 18511101600) and the Natural Science Foundation of Shanghai (Grant No.17ZR1411900 and 14ZR1418500).

**Conflicts of Interest:** We declare that we have no financial and personal relationships with other people or organizations that can inappropriately influence our work, there is no professional or other personal interest of any nature or kind in any product, service and/or company that could be construed as influencing the position presented in, or the review of, the manuscript entitled.

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
