# Peer review of "Spectral Kurtosis of Choi–Williams Distribution and Hidden Markov Model for Gearbox Fault Diagnosis"

_symmetry, doi:10.3390/sym12020285_

Round 1
Reviewer 1 Report
the pictures and the diagrams can be improved in order to be more comprehensive to the reader. the same for the figures' descriptions The experimental setup should be described better the data acquisition points should be clarified. The research and the industrial merit, applicability and contribution of this work's finding should be clarified and highlighted.Author Response
Point 1: The pictures and the diagrams can be improved in order to be more comprehensive to the reader. the same for the figures' descriptions.
Response 1: The pictures and the diagrams have been improved and replaced in the paper.
Point 2: The experimental setup should be described better the data acquisition points should be clarified.
Response 2: The power source of this system is the motor. The input shaft of the first gear box is first connected to the first speed torque sensor, and then connected to the motor. After the first gear box is decelerated, the output shaft and the input of the second gear box. The shaft is connected, the output shaft is connected to the second speed torque sensor after deceleration of the second gear box, and finally connected to the eddy current brake. So that the location of the fault is relatively close, and the required data can be accurately and sensitively collected. Collect vibration signals in five states of gearbox, such as normal, slight fault, moderate fault, severe fault and flaking. The frequency of shaft II is 7Hz. The frequency of shaft I is 10Hz. The sampling points is 4096. The sampling frequency is 3387.77 Hz. The gearbox meshing frequency is 307Hz.
Point 3: The research and the industrial merit, applicability and contribution of this work's finding should be clarified and highlighted.
Response 3: This article monitors the gearbox to master the operating status of the gearbox. When the gearbox is abnormal, it can send an alarm message to the operator in real time so that the fault can be responded to in time. Firstly, the vibration signals of the gearbox are collected in real time. At the same time, time-domain, frequency-domain, and LMD decomposition diagrams can be obtained. Then, an analysis diagram based on CWD-SK is prepared to prepare for the extraction of fault feature vectors. Therefore, when an initial failure occurs, the location of the failure, the degree of the failure can be found, and the faulty component can be replaced or the operating status can be adjusted in time. Finally, HMM training is performed to classify faults. The gearbox is monitored when it has not yet failed, helping operators reduce the development trend of the failure and possible consequences, give maintenance suggestions and countermeasures, and strive to eliminate the failure in the bud state.

Reviewer 2 Report
Literature in the introduction should not be simple listed as a collection of articles, but it should be written in the form of a story. The novelty should be clearly highlighted at the end of the literature.
Acronyms should be explained before being used in the text. Format/style of the paper needs revisiting. In some paragraphs verbs are omitted and punctuation is not used properly. This makes the paper unclear.
Experimental data analysis should be descriptive and not simply list the steps followed unless bullet points are used. The authors mention that from experimental data, the HMM fault diagnosis model can identify the fault states of the gearbox with high accuracy and small amount of data, however, the details on the level of accuracy are not included, nor the amount of needed experimental data.
Author Response
Point 1: Literature in the introduction should not be simple listed as a collection of articles, but it should be written in the form of a story. The novelty should be clearly highlighted at the end of the literature.
Response 1: The introduction has been corrected in the paper. From the discovery of SK to the development process and step-by-step application in fault diagnosis.
Point 2: Acronyms should be explained before being used in the text. Format/style of the paper needs revisiting. In some paragraphs verbs are omitted and punctuation is not used properly. This makes the paper unclear.
Response 2: The acronym has been used for the full name when it was first used. Verbs and punctuation in paragraphs have been corrected in the paper.
Point 3: Experimental data analysis should be descriptive and not simply list the steps followed unless bullet points are used. The authors mention that from experimental data, the HMM fault diagnosis model can identify the fault states of the gearbox with high accuracy and small amount of data, however, the details on the level of accuracy are not included, nor the amount of needed experimental data.
Response 3: For the accuracy of the HMM fault diagnosis model, a table was added to explain its recognition rate. The extracted CWD-SK is used as the feature vector to input HMM and back propagation (BP) neural network to compare the accuracy of fault diagnosis.
Select 20 groups (5 groups of each state) based on the CWD-SK feature vector as training samples.
